# Weakly Supervised Instance Segmentation using the Bounding Box Tightness Prior

Cheng-Chun Hsu[1]*    Kuang-Jui Hsu[2]*    Chung-Chi Tsai[2]
Yen-Yu Lin[1,3]    Yung-Yu Chuang[1,4]
[1]Academia Sinica    [2]Qualcomm Technologies, Inc.
[3]National Chiao Tung University    [4]National Taiwan University
hsu06118@citi.sinica.edu.tw    {kuangjui, chuntsai}@qti.qualcomm.com
lin@cs.nctu.edu.tw    cyy@csie.ntu.edu.tw

## Abstract

This paper presents a weakly supervised instance segmentation method that consumes training data with tight bounding box annotations. The major difficulty lies in the uncertain figure-ground separation within each bounding box since there is no supervisory signal about it. We address the difficulty by formulating the problem as a multiple instance learning (MIL) task, and generate positive and negative bags based on the sweeping lines of each bounding box. The proposed deep model integrates MIL into a fully supervised instance segmentation network, and can be derived by the objective consisting of two terms, i.e., the unary term and the pairwise term. The former estimates the foreground and background areas of each bounding box while the latter maintains the unity of the estimated object masks. The experimental results show that our method performs favorably against existing weakly supervised methods and even surpasses some fully supervised methods for instance segmentation on the PASCAL VOC dataset. The code is available at https://github.com/chengchunhsu/WSIS_BBTP.

## 1   Introduction

Instance-aware semantic segmentation [1, 2, 3, 4], or *instance segmentation* for short, has attracted increasing attention in computer vision. It involves object detection and semantic segmentation, and aims to jointly detect and segment object instances of interest in an image. As a crucial component to image understanding, instance segmentation facilitates many high-level vision applications ranging from autonomous driving [5, 6], pose estimation [7, 8, 9] to image synthesis [10].

Significant progress on instance segmentation such as [7, 11, 12, 13, 14] has been made based on *convolutional neural networks* (CNNs) [15] where instance-aware feature representations and segmentation models are derived simultaneously. Despite effectiveness, these CNN-based methods require a large number of training images with instance-level pixel-wise annotations. Collecting this kind of training data is labor-intensive because of the efforts on delineating the contours of object instances as mentioned in the previous work [16]. The expensive annotation cost has restricted the applicability of instance segmentation.

In this work, we address this issue by proposing a CNN-based method where a model is learned for instance segmentation using training data with bounding box annotations. Compared with the contour, the bounding box of an object instance can be labeled by simply clicking the four outermost points of that instance, leading to a greatly reduced annotation cost. There is only one method for instance segmentation [17] that adopts box-level annotated training data. However, the method [17] is not

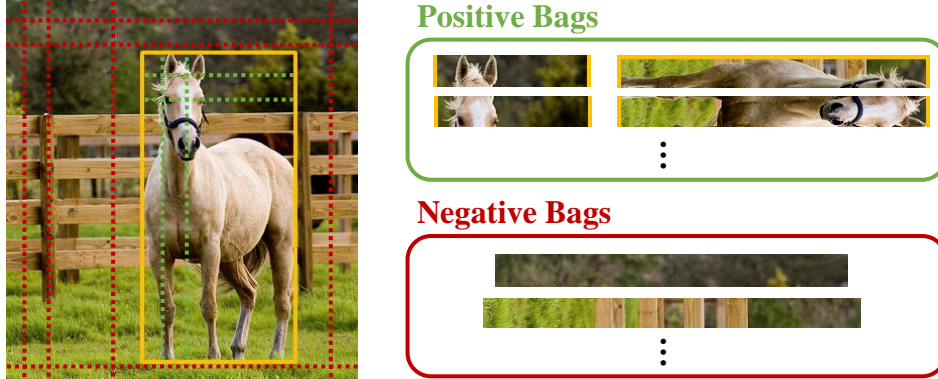

Figure 1: Bounding box annotations, e.g., the yellow rectangle, conform the tightness prior. Namely, each object such as the horse here should touch the four sides of its bounding box. Each *crossing line* whose endpoints are on the opposite sides of a box must contain at lease one pixel belonging to the object. We leverage this property and represent horizontal and vertical crossing lines, e.g., the green ones, as positive bags in multiple instance learning. In contrast, any horizontal and vertical lines, e.g., the red ones, that do not overlap any bounding boxes yield negative bags. Note that, although example bags here are visualized as rectangles, they are for illustrative purpose only. In practice, they are horizontal/vertical lines with 1-pixel heights/widths.

end-to-end trainable. It utilizes GrabCut [18] and MCG proposals [19] to compile pseudo ground truth before learning a fully supervised model for instance segmentation. We deal with the unavailability of ground-truth object instances via *multiple instance learning* (MIL), and integrate the proposed MIL formulation into fully supervised CNN model training. In this way, the instance-aware feature representation and segmentation model are learned by taking the latent ground truth into account. It turns out that the proposed method is end-to-end trainable, resulting in substantial performance improvement. In addition, compared to the existing method [17], our method is proposal-free during inference and considerably speeds up instance segmentation.

Our method is inspired by the tightness prior inferred from bounding boxes [20], as shown in Figure 1. The tightness property states that an object instance should touch all four sides of its bounding box. We leverage this property to design an MIL formulation. Training data in MIL are composed of positive and negative bags. A positive bag contains at least one positive instance while a negative bag contains only negative instances. As shown in Figure 1, a vertical or a horizontal *crossing line* within a bounding box yields a positive bag because it must cover at least one pixel belonging to the object. A horizontal or vertical line that does not pass through any bounding boxes forms a negative bag. We integrate the MIL formulation into an instance segmentation network, and carry out end-to-end learning of an instance segmentation model with box-level annotated training data.

The major contributions of this work are summarized as follows. First, we present a CNN-based model for weakly supervised instance segmentation. The task of inferring the object from its bounding box is formulated via MIL in our method. To the best of our knowledge, the proposed method offers the first end-to-end trainable algorithm that learns the instance segmentation model using training data with bounding box annotations. Second, we develop the MIL formulation by leveraging the tightness property of bounding boxes, and incorporate it in instance segmentation. In this way, latent pixel-wise ground truth, the object instance feature representation, and the segmentation model can be derived simultaneously. Third, existing instance segmentation algorithms work based on the detected object bounding box. The quality of detected object boxes limits the performance of instance segmentation. We address this issue by using DenseCRF [21] to refine the instance masks. Finally, when evaluated on the popular instance segmentation benchmark, the PASCAL VOC 2012 dataset [22], our method achieves substantially better performance than the state-of-the-art box-level instance segmentation method [17].

## 2 Related Work

**Weakly supervised semantic segmentation.** CNNs [15] have demonstrated the effectiveness for joint feature extraction and non-linear classifier learning. The state-of-the-art semantic segmentation

methods [23, 24, 25, 26, 27, 28, 29, 30, 31] are developed based on CNNs. Owing to the large capability of CNNs, a vast amount of training data with pixel-wise annotations are required for learning a semantic segmentation model without overfitting. To address this issue, different types of weak annotations have been adopted to save manual efforts for training data labeling, such as synthetic annotations [32], bounding boxes [33, 17, 34], scribble [35, 36, 37, 38], points [39, 40], and image-level supervision [41, 42, 43, 44, 45, 46, 47, 48, 49].

Our method adopts bounding box annotations that come to a compromise between the instance segmentation accuracy and the annotation cost. Different from existing methods [33, 17, 34] that also use box-level annotations, our method targets at the more challenging instance segmentation problem rather than semantic segmentation. In addition, our method makes the most of bounding boxes by exploring their tightness properties to improve instance segmentation.

**Fully supervised instance segmentation.** SDS [1] extends the R-CNN [50] scheme for instance segmentation by jointly considering the box proposals and segment proposals pre-generated by selective search [51]. The early methods, such as [2, 3, 4], follow this strategy for instance segmentation. However, their performance heavily depends on the quality of the pre-generated proposals. Furthermore, extracting region proposals [51] is typically computationally intensive. The *region proposal network* (RPN) in Faster R-CNN [52] offers an efficient way to generate high-quality proposals. Recent instance segmentation methods [53, 54, 7, 11, 12, 13, 14] benefit from RPN and an object detector like Faster R-CNN. These methods first detect object bounding boxes and perform instance segment on each detected bounding box. Another branch for instance segmentation is to segment each instance directly without referring to the detection results, such as [55, 56, 57, 58, 59, 5, 60]. Methods of this category rely on pixel-wise annotated training data, which may restrict their applicability.

**Weakly supervised instance segmentation.** There exist few deep-learning-based methods that use weak annotations for instance segmentation, such as box-level annotations [17], image-level annotations [61], and image groups [62]. Khoreva *et al.*'s method [17] is the first and only CNN-based method that consumes box-level supervisory training data for instance segmentation. However, their two-stage method is not end-to-end trainable. In its first stage, instance-level pixel-wise pseudo ground truth is generated using GrabCut [18] and the MCG proposals [19]. In the second stage, the generated pseudo ground truth is used to train their instance segmentation algorithm called DeepLab$_{box}$ to complete instance segmentation. Thus, the performance of their method [17] is bounded by the pseudo ground truth generators, i.e., GrabCut [18] and the MCG proposals [19]. Different from [17], the methods in [61, 62] respectively utilize image-level supervision and self-supervised learning within an image group to carry out weakly supervised instance segmentation. In contrast, our method handles the uncertainty of inferring object segments from their bounding boxes via the proposed MIL formulation. The proposed method is end-to-end trainable so that pseudo ground truth estimation, instance feature representation learning, and segmentation model derivation can mutually refer to and facilitate each other during training. Furthermore, our method does not require object proposals during inference. As a result, the proposed method performs favorably against the competing method [17] in both accuracy and efficiency. Some frameworks [63, 64, 65] use training data with a mixture of few pixel-wise annotations and abundant box-level annotations. In contrast, our framework does not rely on pixel-wise annotations.

# 3 Proposed Method

The proposed method is introduced in this section. We first give an overview to the method, and then describe the proposed MIL formulation followed by specifying the objective for network optimization. Finally, the adopted segment refinement techniques and the implementation details are provided.

## 3.1 Overview

We are given a set of training data with bounding box annotations, $\mathcal{D} = \{I_n, B_n\}_{n=1}^N$ where $N$ is the number of images, $I_n$ is the $n$th image, and $B_n$ is the box-level annotations for $I_n$. Suppose the image $I_n$ contains $K_n$ bounding boxes. Its annotations would be in the form of $B_n = \{\mathbf{b}_n^k, \mathbf{y}_n^k\}_{k=1}^{K_n}$ where the location label $\mathbf{b}_n^k$ is a 4-dimensional vector representing the location and size of the $k$th box, the class label $\mathbf{y}_n^k$ is a $C$-dimensional vector specifying the category of the corresponding box, and $C$ is the number of classes. In this work, we aim to learn a CNN-based model in an end-to-end fashion for instance segmentation by using the training set $\mathcal{D}$.

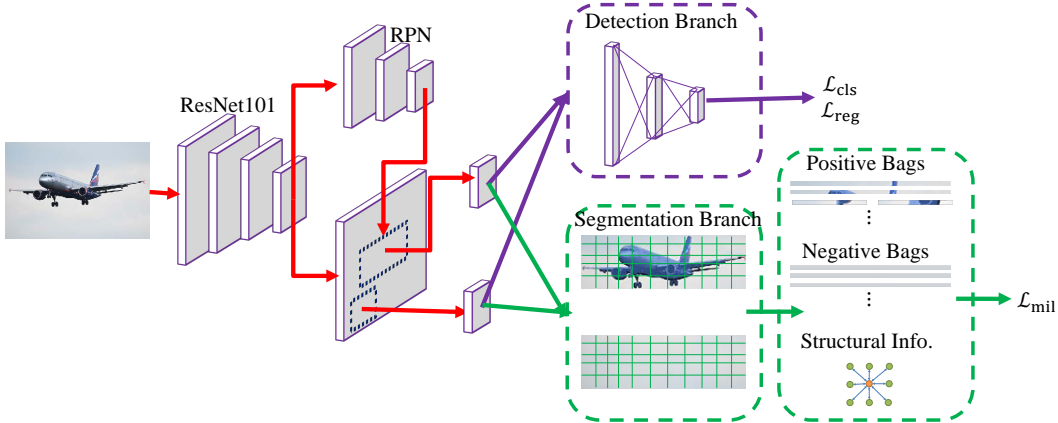

Figure 2: Overview of our model, which has two branches, the detection and segmentation branches. The Mask R-CNN framework is adopted. First, we extract the features with ResNet 101, and then use RPN and ROI align to generate the region feature for each detected bounding box. The detection branch consists of the fully connected layers and two losses, the box classification loss, $\mathcal{L}_{\text{cls}}$, and the box regression loss, $\mathcal{L}_{\text{box}}$. In the segmentation branch, we first estimate the object instance map inside each detected bounding box, and then generate the positive and negative bags using the bounding box tightness prior for MIL. The MIL loss, $\mathcal{L}_{\text{mil}}$, is optimized with the bags and the additional structural constraint. The model can be jointly optimized by these three losses in an end-to-end manner only with the box-level annotations.

The overview of our method is given in Figure 2 where the proposed MIL formulation is integrated into a backbone network for fully supervised instance segmentation. In this work, we choose Mask R-CNN [7] as the backbone network owing to its effectiveness though our method is flexible to team up with any existing fully supervised instance segmentation models. Learning the Mask R-CNN [7] model requires instance-level pixel-wise annotations. To save the annotation cost and fit the problem setting, we adopt the bounding box tightness prior for handling weakly annotated training data, and formulate it as an MIL objective to derive the network. The tightness prior is utilized to augment the training data $\mathcal{D}$ with a set of bag-structured data for MIL. With the augmented training data, the network for instance segmentation can be optimized using the following loss function:

$$\mathcal{L}(\mathbf{w}) = \mathcal{L}_{\text{cls}}(\mathbf{w}) + \mathcal{L}_{\text{reg}}(\mathbf{w}) + \mathcal{L}_{\text{mil}}(\mathbf{w}), \tag{1}$$

where $\mathcal{L}_{\text{cls}}$ and $\mathcal{L}_{\text{reg}}$ are the box classification and regression losses, respectively, and $\mathcal{L}_{\text{mil}}$ is the proposed MIL loss. In Eq. (1), the classification loss $\mathcal{L}_{\text{cls}}$ assesses the accuracy of the box classification task while the regression loss $\mathcal{L}_{\text{reg}}$ measures the goodness of the box coordinate regression. Both $\mathcal{L}_{\text{cls}}$ and $\mathcal{L}_{\text{reg}}$ can be directly optimized using the box-level annotations. We implement $\mathcal{L}_{\text{cls}}$ and $\mathcal{L}_{\text{reg}}$ as proposed in [7], and omit the details here. The proposed loss $\mathcal{L}_{\text{mil}}$ enables end-to-end network optimization by using training data with box-level annotations instead of the original pixel-level ones. The details of $\mathcal{L}_{\text{mil}}$ are provided in the following.

### 3.2 Proposed MIL formulation

We describe how to leverage the tightness prior inferred from bounding boxes to yield the bag-instance data structure for MIL. The bounding box of an object instance is the smallest rectangle enclosing the whole instance. Thus, the instance must touch the four sides of its bounding box, and there is no overlap between the instance and the region outside its bounding box. These two properties will be used to construct the positive and negative bags respectively in this work.

A crossing line of a bounding box is a line with its two endpoints locating on the opposite sides of the box. Pixels on a crossing line compose a positive bag of the corresponding box category, since the line has at least one pixel belonging to the object within the box. On the contrary, pixels on a line that does not pass through any bounding boxes of the category yield a negative bag because no object of that category is present outside the bounding boxes. In this work, for each bounding box, we collect all horizontal and vertical crossing lines to produce the positive bags. A positive bag can be denoted as $\hat{b}^{+} = \{\mathbf{p}_i\}$, where $\mathbf{p}_i$ is the $i$th pixel on the line. We also randomly sample the same number of

negative bags, each of which corresponds to a line near and outside the bounding box. Similarly, a negative bag is expressed as $\hat{b}^- = \{\mathbf{p}_i\}$.

We augment the given training set $\mathcal{D} = \{I_n, B_n\}_{n=1}^N$ with the generated bag-structured data. The resultant augmented dataset is denoted as $\hat{\mathcal{D}} = \{I_n, B_n, \tilde{B}_n\}_{n=1}^N$, where $\tilde{B}_n = \{\hat{B}_{n,k}^+, \hat{B}_{n,k}^-\}_{k=1}^{K_n}$ contains all positive and negative bags of the $k$th bounding box $\mathbf{b}_n^k$ in image $I_n$. Specifically, the positive set $\hat{B}_{n,k}^+ = \{\hat{b}_{n,k,\ell}^+\}_{\ell=1}^{H_{n,k}+W_{n,k}}$ consists of $H_{n,k} + W_{n,k}$ bags, each of which corresponds to a crossing vertical or horizontal line within the bounding box, where $H_{n,k}$ and $W_{n,k}$ are the height and width of the bounding box $\mathbf{b}_n^k$, respectively. Similarly, we have the set of negative bags $\hat{B}_{n,k}^- = \{\hat{b}_{n,k,\ell}^-\}_{\ell=1}^{H_{n,k}+W_{n,k}}$.

## 3.3 MIL loss

This section specifies the design of the loss function $\mathcal{L}_{\text{mil}}$ with the augmented dataset $\hat{\mathcal{D}}$. As shown in Figure 2, given the augmented training set $\hat{\mathcal{D}}$, the detection branch of the network can be optimized by using the ground-truth bounding boxes through the two loss functions $\mathcal{L}_{\text{cls}}$ and $\mathcal{L}_{\text{reg}}$. On the other hand, the segmentation branch predicts an instance score map $S_{n,k} \in [0,1]^{W_{n,k} \times H_{n,k}}$ for each bounding box $\mathbf{b}_n^k$ with respect to its object category. To train this branch, we develop the loss function $\mathcal{L}_{\text{mil}}$ based on MIL and the augmented bag-structured data. Considering each bounding box $\mathbf{b}$, we use its corresponding sets of the positive and negative bags, $\hat{B}^+$ and $\hat{B}^-$, respectively. We omit the image and box indices here for the sake of brevity. The MIL loss $\mathcal{L}_{\text{mil}}$ consists of two terms as defined by

$$\mathcal{L}_{\text{mil}}(S; \hat{B}^+, \hat{B}^-) = \psi(S; \hat{B}^+, \hat{B}^-) + \phi(S), \tag{2}$$

where the unary term $\psi$ enables MIL by enforcing the tightness constraints with the training bags $\hat{B}^+$ and $\hat{B}^-$ on $S$, and the pairwise term $\phi$ imposes the structural constraint on $S$ for maintaining integrity of the object. The unary term $\psi$ and the pairwise term $\phi$ in Eq. (2) are described below.

**Unary term.** Given the sets of positive and negative bags $\hat{B}^+$ and $\hat{B}^-$, the unary term enforces the tightness constraints of the bounding boxes on the prediction map $S$. It also helps the network to predict better instance masks. As discussed in Section 3.2, a positive bag must contain at least one pixel inside the object segment, so we encourage the maximal prediction score among all pixels in a positive bag to be as close to 1 as possible. In contrast, no pixels in a negative bag belong to an object of the category, and hence we minimize the maximal prediction score among all pixels in a negative bag. We implement the two observations by defining the unary term as

$$\psi(S; \hat{B}^+, \hat{B}^-) = \sum_{\hat{b} \in \hat{B}^+} -\log P(\hat{b}) + \sum_{\hat{b} \in \hat{B}^-} -\log\left(1 - P(\hat{b})\right), \tag{3}$$

where $P(\hat{b}) = \max_{\mathbf{p} \in \hat{b}} S(\mathbf{p})$ is the estimated probability of the bag $\hat{b}$ being positive, and $S(\mathbf{p})$ is the score value of the map $S$ at position $\mathbf{p}$. The probability $P(\hat{b})$ in Eq. (3) can be efficiently computed by column- and row-wise maximum pooling, which can be implemented easily without changing the network architecture.

**Pairwise term.** Using the unary term alone is prone to segment merely the discriminative parts of an object rather than the whole object, as pointed out in previous work [66, 67]. The pairwise term addresses this issue and uses the structural constraint to enforce the piece-wise smoothness in the predicted instance masks. In this way, the high scores of the discriminative parts can be propagated to its neighborhood. We explicitly model the structural information by defining the following pairwise term:

$$\phi(S) = \sum_{(\mathbf{p}, \mathbf{p}') \in \varepsilon} \|S(\mathbf{p}) - S(\mathbf{p}')\|^2, \tag{4}$$

where $\varepsilon$ is the set containing all neighboring pixel pairs.

The proposed MIL loss in Eq. (2) is differentiable and convex, so our network can be efficiently optimized by stochastic gradient descent. The gradients of all loss functions in Eq. (1) with respect to the optimization variables can be derived straightforwardly, and hence are omitted here.

| method | publication | Sup. | mAP$^r_{0.25}$ | mAP$^r_{0.5}$ | mAP$^r_{0.7}$ | mAP$^r_{0.75}$ |
|---|---|---|---|---|---|---|
| M. R-CNN [7] | ICCV'17 | M | 76.7 | 67.9 | 52.5 | 44.9 |
| SDS [1] | ECCV'14 | M | - | 49.7 | 25.3 | - |
| PRM [61] | CVPR'18 | I | 44.3 | 26.8 | - | 9.0 |
| DetMCG | - | B | 38.3 | 19.6 | 8.2 | 5.4 |
| BoxMask | - | B | 69.2 | 34.5 | 10.0 | 6.7 |
| BoxMCG | - | B | 53.1 | 37.1 | 19.8 | 14.7 |
| SDI [17] | CVPR'17 | B | - | 44.8 | - | 16.3 |
| Ours | - | B | 75.0 | 58.9 | 30.4 | 21.6 |
| M. R-CNN [7] | ICCV'17 | M* | 74.8 | 67.5 | 54.1 | 47.9 |
| SDI [17] | CVPR'17 | B* | - | 46.4 | - | 18.5 |
| Ours | - | B* | 77.2 | 60.1 | 29.4 | 21.2 |

Table 1: Evaluation of instance segmentation results from different methods. The field "Sup." indicates the supervision type, $I$ for image-level labels, $B$ for box-level labels, and $M$ for mask-level labels. An asterisk * indicates the use of the training set containing both the MS COCO and PASCAL VOC 2012 datasets.

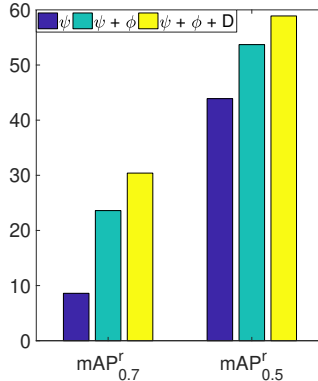

Figure 3: The ablation study for component contributions on the PASCAL VOC 2012 dataset.

## 3.4 DenseCRF for object instance refinement

A major limitation of detection-based methods for instance segmentation is that they highly suffer from the problem caused by inaccurately detected object boxes, since instance segmentation is based on the detected bounding boxes. To address this issue, we use DenseCRF [21] for refinement. During inference, for a detected bounding box, we first generate the score map for the detected box using the trained segmentation branch. Then, the predicted score map of the box is pasted to a map $\hat{S}$ of the same size as the input image according to the box's location. For setting up DenseCRF, we employ the map $\hat{S}$ as the unary term and use the color and pixel location differences with the bilateral kernel to construct the pairwise term. After optimization using mean field approximation, DenseCRF produces the final instance mask. As shown in the experiments, DenseCRF compensates for the inaccurate detection by the object detector.

## 3.5 Implementation details

We implement the proposed method using *PyTorch*. ResNet-101 [68] serves as the backbone network. It is pre-trained on the ImageNet dataset [69], and is updated during optimizing Eq. (1). The same network architecture is used in all experiments. During training, the network is optimized on a machine with four GeForce GTX 1080 Ti GPUs. The batch size, learning rate, weight decay, momentum, and the number of the iterations are set to 8, $10^{-2}$, $10^{-4}$, 0.9 and 22k, respectively. We choose ADAM [70] as the optimization solver because of its fast convergence. For data augmentation, following the setting used in Mask R-CNN, we horizontally flip each image with probability 0.5, and randomly resize each image so that the shorter side is larger than 800 pixels and the longer side is smaller than 1,333 pixels, while maintaining the original aspect ratio.

## 4 Experimental Results

The proposed method is evaluated in this section. First, we describe the adopted dataset and evaluation metrics. Then, the performance of the proposed method and the competing methods is compared and analyzed. Finally, the ablation studies on each proposed component and several baseline variants are conducted.

### 4.1 Dataset and evaluation metrics

**Dataset.** The Pascal VOC 2012 [22] dataset is widely used in the literature of instance segmentation [1, 17, 61]. This dataset consists of 20 object classes. Following the previous work [17], we use the augmented Pascal VOC 2012 dataset [71] which contains totally 10,582 training images.

---

The authors from Academia Sinica and the universities in Taiwan completed the experiments on the datasets.

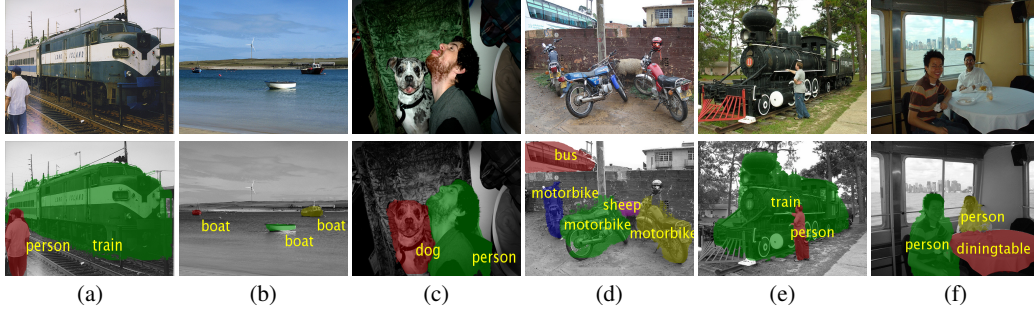

|  (a) | (b) | (c) | (d) | (e) | (f) |

Figure 4: The examples of the segmentation results with our method. The top row are the input images while the bottom row shows the corresponding segmentation results. In the segmentation results, different instances are indicated by different colors. Their categories are identified by texts.

In addition, for fair comparison with the SDI method [17], we also train our method using the additional training images from the MS COCO [72] dataset and report the performance. Following the same setting adopted in SDI [17], we only select the images with objects whose categories are covered by the Pascal VOC 2012 dataset and bounding box areas larger than 200 pixels. The number of the selected images is $99,310$. We add the selected images to the training set. Note that only box-level annotations are used in all the experiments.

**Evaluation metrics.** The standard evaluation metric, the mean average precision (mAP) [1], is adopted. Following the same evaluation protocol in [17, 61], we report mAP with four IoU (intersection over union) thresholds, including $0.25, 0.5, 0.7$, and $0.75$, denoted as $\text{mAP}_k^r$ where $k \in \{0.25, 0.5, 0.7, 0.75\}$.

## 4.2 Comparison with the state-of-the-art methods

We compare the proposed method with several state-of-the-art methods including PRM [61], SDI [17], Mask R-CNN [7] and SDS [1]. The results are reported in Table 1, where the field "Sup." indicates the supervision types, $I$ for image-level labels, $B$ for box-level labels and $M$ for mask-level labels. In addition, the asterisk $^*$ in the field "Sup." indicates the use of the mixed training set which combines the MS COCO and PASCAL VOC 2012 datasets. Among the compared methods, Mask R-CNN [7] and SDS [1] are fully supervised with object masks. SDI [17] and PRM [61] are weakly supervised. The supervision type of SDI is the same as our method with box-level annotations, while PRM [61] uses only image-level annotations.

As shown in Table 1, our proposed method significantly outperforms SDI [17] by large margins, around $14.1\%$ in $\text{mAP}_{0.5}^r$ and $5.3\%$ in $\text{mAP}_{0.75}^r$ on the Pascal VOC 2012 dataset. Noted that SDI [17] is the state-of-the-art instance segmentation method using box-level supervision. When training on both the MS COCO and Pascal VOC 2012 datasets, our method also outperforms SDI by large margins, around $13.7\%$ in $\text{mAP}_{0.5}^r$ and $2.7\%$ in $\text{mAP}_{0.75}^r$. It is also worth mentioning that our method even achieves a remarkably better performance than the fully supervised method, SDS [1]. Thus, we believe that the proposed MIL loss is really useful to the box-level supervision setting for instance segmentation and also likely benefits other tasks which require instance masks or use box-level annotations. In addition, since we use the Mask R-CNN as the backbone, training it with the ground truth masks provides the upper bound of our method. From Table 1, our method achieves comparable performance with Mask R-CNN in terms of $\text{mAP}_{0.5}^r$, showing that the proposed method method is effective on utilizing the information provided by bounding boxes. Note that, our method adopting the MIL formulation tends to highlight the discriminative parts of objects, while Mask R-CNN with mask-level annotations emphasizes the whole objects. The IOUs between the ground truth and the discriminative regions are often larger than $0.25$ but less than $0.5$. It is why our method slightly outperforms Mask R-CNN in $\text{mAP}_{0.25}^r$, but falls behind in others.

Figure 4 shows some instance segmentation examples using the proposed method. Our approach can produce high-quality results even in challenging scenarios. All examples in Figure 4 except (b) exhibit occlusions between instances. Our method can distinguish instances even when they are closed or occluded with each other. Examples in (a), (b) and (e) shows that our method performs

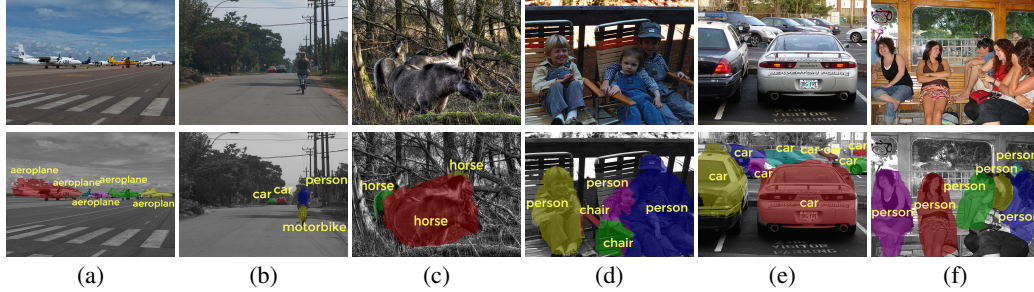

|       (a)       |       (b)       |       (c)       |       (d)       |       (e)       |       (f)       |

Figure 5: The failure examples produced by our proposed method.

well with objects of different scales. In (d), there are objects of several classes and complex shapes. Our method can segment multi-class instances with the complex shapes very well, showing that the proposed method can delineate fine-detailed object structures. Some failure cases are shown in Figure 5. As the proposed method is a weakly-supervised method, it may be misled by noisy co-occurrence patterns and have problems when separating different object instances of the same class. For example, in (a) and (b), segments of small objects are incomplete due to the unclear boundaries in low-resolution regions. In (c) and (d), different instances are wrongly merged. In (e) and (f), inaccurate object contours are segmented due to inter-instance similarity and cluttered scenes.

## 4.3 Ablation studies

We conduct four types of ablation studies, including the baseline studies, the analysis of the contribution of each proposed component, the robustness to inaccurate annotations, and the performance with different annotation costs.

### 4.3.1 Baseline studies

For better investigating how well the proposed method utilizes box-level annotations, we construct three baseline methods which also perform instance segmentation using box-level annotations. As our model, all of them are based on Mask R-CNN. The first baseline, *DetMCG*, trains Mask R-CNN without the segmentation branch and the network only detects bounding boxes of objects. During inference, given a detected bounding box, we retrieve the MCG proposal with the highest IoU with the detected box as the output instance mask. The second baseline, *BoxMask*, trains Mask R-CNN by regarding the ground-truth boxes as the object instance masks. In the third baseline, *BoxMCG*, for each ground-truth bounding box, we retrieve the MCG proposal with the highest IoU with it and regard the proposal as the pixel-level annotation for training Mask R-CNN. Table 1 reports the performance of the three baseline methods. Their performance is much worse than the proposed method, showing that our method utilizes the box-level annotations much more effectively.

For *DetMCG*, the information of the object inside the bounding box is not explored, and the detection and segment branches are not jointly optimized. *BoxMask* uses bounding boxes as masks. It completely ignores the fine object contours and regards many background pixels as objects during training, which can be misleading. In *BoxMCG*, although object contours are considered, inaccurate object proposals could contain many false positives and false negatives and hurt the performance when serving as the training masks. Therefore, it can only achieve sub-optimal results. In contract, in the proposed method, the detection and segmentation branches are jointly trained, while the tightness property is utilized for distinguishing the foreground pixels from background pixels. In addition, spatial coherence is enforced to better maintain the integrity of objects. Therefore, our method can achieve much better results compared to the naive baselines.

### 4.3.2 Component contributions

We analyze the contributions of each proposed components, including terms in the proposed MIL objective (Eq. (2)) and DenseCRF, on the Pascal VOC 2012 dataset. The results of $\text{mAP}^r_{0.5}$ and $\text{mAP}^r_{0.7}$ are reported in Figure 3. First, for the MIL loss in Eq. (2), the performance gain by adding the pairwise term $\phi$ is significant in both measures. It is because the term helps better discover the whole objects instead of focusing only on their discriminative parts. DenseCRF can moderately improve the performance by correcting inaccurate box detection and refining the object shapes.

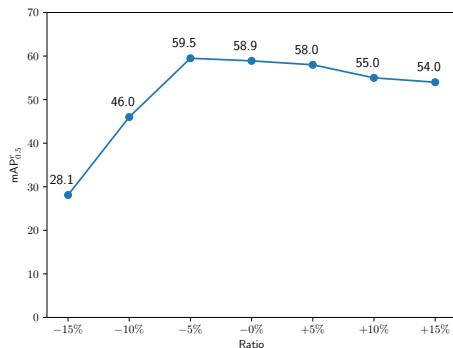
Figure 6: Robustness to inaccurate annotations.

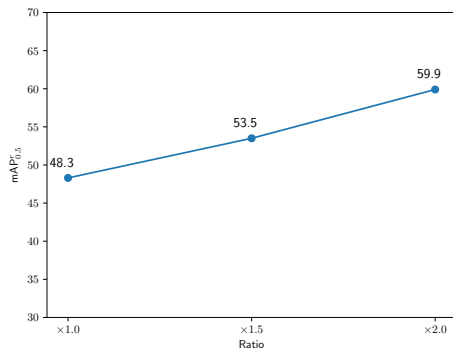
Figure 7: Study of annotation costs.

## 4.4 Robustness to inaccurate annotations

We study the robustness and sensitivity of the proposed method to the accuracy of bounding boxes by expanding/contracting the boxes and evaluating the performance under different expansion/contraction ratios on the Pascal VOC 2012 dataset. Figure 6 shows the results. Noted that all the results are evaluated in $mAP_{0.5}^r$. The proposed method is quite robust to small change ratios from $-5\%$ to $+5\%$. In fact, by slightly contracting the bounding boxes, e.g. $-5\%$, we can reduce noisy pixels in each positive bag, resulting in even better performance. However, excessively expanding or contracting ratios leads to unreliable positive and negative bags, and thus produces sub-optimal results.

## 4.5 Performance versus different annotation costs

According to [73], the instance-level and box-level annotation costs on the Pascal VOC 2012 dataset are 239.7 and 38.1 seconds per image, respectively. We train Mask R-CNN by using instance-level annotations, and limit the amount of annotations so that the annotation budget is comparable with $1.0\times$, $1.5\times$, and $2.0\times$ of the box-level annotation cost, respectively. As shown in Figure 7, the results of the three different settings on the Pascal VOC 2012 dataset are $48.3\%$, $53.5\%$ and $59.9\%$ in $mAP_{0.5}^r$, respectively. The first two results fall behind our method by the margins $10.6\%$ and $5.4\%$ respectively, while the last one surpasses ours by $1.0\%$ but with $2\times$ of the annotation cost. Therefore, with the same annotation cost, our method outperforms Mask R-CNN because less training data lead to overfitting for Mask R-CNN.

## 5 Conclusion

In this paper, we propose a weakly supervised instance segmentation method, which can be trained with only box-level annotations. For achieving figure-ground separation using only information provided by bounding boxes, we integrate the MIL formulation into a fully supervised instance segmentation network. We explore the tightness prior of the bounding boxes for effectively generating the positive and negative bags for MIL. By integrating spatial coherence and DenseCRF, the integrity and shape of the object can be better preserved. Experiments show that the proposed method outperforms existing weakly supervised methods and even surpasses some fully supervised methods for instance segmentation on the PASCAL VOC 2012 dataset. With the simpler box-level annotations, the proposed method could expand the utility of instance segmentation. In addition, we believe that the proposed scheme can also benefit other tasks using box-level annotations.

### Acknowledgments

This work was funded in part by Qualcomm through a Taiwan University Research Collaboration Project and also supported in part by the Ministry of Science and Technology (MOST) under grants 107-2628-E-001-005-MY3 and 108-2634-F-007-009, and MOST Joint Research Center for AI Technology and All Vista Healthcare under grant 108-2634-F-002-004.

## Footnotes

*indicates equal contributios

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
