[Supplementary Material]

# Supplementary material: Weakly Supervised Instance Segmentation using the Bounding Box Tightness Prior

In this supplementary material, we provide more detailed analysis of our proposed methods, including class-wise results and more visualization examples of our predicted results on the PASCAL VOC 2012 dataset.

## 1 Class-wise Results

| | Avg. | A.P. | Bike. | Bird | Boat | Bottle. | Bus | Car | Cat | Chair | Cow | D.T. | Dog | Horse | M.B. | P.S. | P.P. | Sheep | Sofa | Train | TV |
|---|---|---|---|---|---|---|---|---|---|---|---|---|---|---|---|---|---|---|---|---|---|
| $\text{mAP}_{0.25}^r$ | 75.0 | 87.7 | 19.1 | 87.8 | 71.1 | 62.0 | 87.7 | 74.8 | 94.3 | 43.4 | 83.3 | 65.0 | 94.4 | 84.7 | 89.4 | 82.7 | 61.8 | 72.9 | 63.3 | 91.4 | 82.5 |
| $\text{mAP}_{0.50}^r$ | 58.9. | 63.8 | 0.3 | 69.2 | 44.7 | 51.9 | 83.6 | 63.0 | 92.4 | 14.3 | 72.0 | 37.2 | 86.0 | 49.6 | 78.9 | 62.5 | 42.7 | 57.6 | 50.2 | 84.3 | 74.0 |
| $\text{mAP}_{0.70}^r$ | 30.4 | 22.3 | 0.0 | 21.0 | 21.0 | 31.1 | 70.3 | 40.5 | 65.4 | 5.4 | 19.2 | 22.9 | 48.1 | 5.1 | 24.9 | 20.6 | 13.9 | 15.7 | 37.9 | 66.9 | 56.3 |
| $\text{mAP}_{0.75}^r$ | 21.6 | 13.6 | 0.0 | 8.9 | 11.8 | 21.1 | 62.3 | 29.8 | 41.7 | 3.1 | 10.2 | 15.3 | 35.1 | 1.8 | 11.4 | 12.2 | 7.7 | 8.5 | 34.0 | 57.0 | 45.7 |

Table 1: Class-wise performance in the four measures, $\text{mAP}_k^r$ where $k \in \{0.25, 0.5, 0.7, 0.75\}$, of our method for instance segmentation on the PASCAL VOC 2012 dataset.

## 2 Result Visualization

More example results of our proposed method on the PASCAL VOC 2012 dataset are shown in Figure 1, Figure 2, Figure 3 and Figure 4. Our proposed method is robust enough to deal with several difficult variations, such as occlusions, multiple scale or closeness between different instances.

From Figure 1, our model can successfully detect multiple instances and their corresponding masks. For example, in Figure 1 (a), although two instances from different classes, i.e., persons and motorbikes, are presented, the instance segmentation masks are well predicted. Furthermore, in Figure 1 (d) and (f), object occlusions exist. Some parts of the train is occluded by the person in Figure 1 (d) while a person is occluded by a motorbike in Figure 1 (f). However, our model can still predict promising results. Small objects and object closeness are presented in Figure 1 (c) and (e), and in Figure 1, respectively, but the instance still can be well detected and segmented.

In Figure 2, the instances in each images belong to same category, so the detector are more likely to be misled by similar appearance. However, as shown in Figure 2, our model can well distinguish the instances even if they have similar appearance to each other. Moreover, the similar variations discussed in the previous paragraph are also observed in Figure 2, but our method can still predict the promising results.

In Figure 3 and Figure 4, we show some simple cases which only one instance is presented in each image. There exists no occlusion or closeness between the instances, so the predicted results are much better than the ones in Figure 1 and Figure 2.

Figure 1: The examples of the segmentation results produced by our method. From the top row to the bottom row, the input images, the corresponding ground truth masks and the corresponding segmentation results are shown, respectively. In the segmentation results, different instances are indicated by different colors. Their categories are identified by texts.

Figure 2: The examples of the segmentation results produced by our method. From the top row to the bottom row, the input images, the corresponding ground truth masks and the corresponding segmentation results are shown, respectively. In the segmentation results, different instances are indicated by different colors. Their categories are identified by texts.

Figure 3: The examples of the segmentation results produced by our method. From the top row to the bottom row, the input images, the corresponding ground truth masks and the corresponding segmentation results are shown, respectively. In the segmentation results, different instances are indicated by different colors. Their categories are identified by texts.

Figure 4: The examples of the segmentation results produced by our method. From the top row to the bottom row, the input images, the corresponding ground truth masks and the corresponding segmentation results are shown, respectively. In the segmentation results, different instances are indicated by different colors. Their categories are identified by texts.