[Reviews · NeurIPS 2019]

Reviewer 1



The proposed method for performing weakly supervised instance segmentation is an original and novel contribution, extending the Mask R-CNN with a MIL loss term to enable learning of instance segmentation from only bounding box labels. As far as quality, the experimental validation is thorough, including comparisons to two state-of-the-art baselines in weakly supervised instance segmentation as well as comparisons/upper bounds of fully-supervised instance segmentation methods. Additionally, ablation studies are given to validate the use of box-level annotations compared with alternative baseline approaches, as well as to validate the different components of the overall proposed pipeline. The paper is also well-written and easy to read. Overall, given the above, this is a nice paper with significance to anyone working on object detection/segmentation. --- After reading the author response, I believe they have addressed both my questions and concerns raised by the other reviewers, and maintain my initial rating. One comment based on reviewer discussion: In my initial read, Figure 1 was slightly confusing due to the illustration of positive and negative bags as being patches with greater than 1 pixel height/width. Perhaps it would be worth adding a small note clarifying that the example bags are just for illustrative purposes, and that in practice single pixel horizontal/vertical lines are used as bags?

Reviewer 2



This paper is well-organized and the presentation is nice. The ideas are interesting, which I believe could be helpful for future research in weakly-supervised segmentation. The results are also good and convinving. Overall speaking, this paper is well-writen. However, I still have the following questions. 1) The authors use max pooling with respect to each bag to get the probability of each bag belonging to some category. I want to know how the results change when the avg pooling is used. We know that recent classification networks mostly exploit 2D avg pooling to map a 3D tensor to a 1D feature vector. According to attention models (like CAM, CVPR'16), avg pooling is more effective than max pooling to capture large-range object attentions. 2) For objects with middle or large size (defined in COCO), this work may work well. My question is how is the result of the proposed approach when processing small objects.

Reviewer 3



The major issues of this paper are related to the motivation. Though it claims in L50 that this is the first end-to-end trainable algorithm that learns the instance segmentation model using bounding box annotations, this does not explain well the value of such problem. If the motivation of using bounding box annotation for training instance segmentation is that such bounding box is cheaper than boundary annotation, then there should be a study of performance versus annotation effort, e.g., in terms of annotation expense or total annotation time. This will answer if weakly supervised instance segmentation achieves better performance than fully supervised on given the same amount of annotation time/money. It may also be possible that given the same amount of time/money, the fine pixel annotation is better than coarse bounding box annotation in terms of training. [R1] performs such a study as reference in terms of semantic segmentation. [R1] On the Importance of Label Quality for Semantic Segmentation, CVPR, 2018 Line85: This is a misleading statement. For some real-world applications, the pixel-level annotation is required. Moreover, it should be noted that the non-proposal based instance segmentation can be exclusively applied to some other more challenging tasks, like C. elegans segmentation which are more deformable and often cuddle with each other for instance segmentation. In such cases, proposal based methods cannot handle well, though they perform well for segmenting oval-shape objects. Eq.4 How to set epsilon? Why using Eq. 4 helps enlarge the segment size? Why not training to classify all positive patches as positive labels? Why must it use MIL loss given that all the positive patches are positive in some sense? Line100: There is no support for "efficiency" of the proposed method over [16]. In general, the paper about using MIL for weakly supervised instance segmentation is not persuasive. It does not explain well why MIL works so well -- on some metrics it even outperforms the fully-supervised Mask RCNN. Given the results, an in-depth analysis is required to explain the advantage. ------------------------- The rebuttal provides answers to most of my questions. There are still a few concerns -- 1) I still have difficulty in understanding why the MIL works so well with multiple sampled patches as positive/negative bags. From Fig. 1 and Eq. 3, I don't know how the MIL forces the model to choose what patch for the positive bag. Perhaps a visual demonstration may demonstrate this. The authors partially answer this in "The four questions about Eq. 4". 2) The answer provided in the rebuttal is very important that studies performance vs. annotation effort (time, money). That's one of the main motivation why weakly supervised learning is important -- if one has infinite money, then annotation for fully supervised learning is no problem. If this paper is finally accepted, I strongly suggest authors include this in the main paper especially given that the paper is submitted to machine learning venue and includes little theories. Another minor concern is the claim about L85 "in general, the applicability of the fully supervised methods “may” be limited in the real world because of the high annotation cost". This really has ambiguity in defining the "general application" -- production oriented application in companies, or numerous applications in-need in biology/medical science, or others. That's why I would rather see the performance vs. annotation cost to motivate weakly supervised learning instead of saying this vague statement.

[Author Response · NeurIPS 2019]



Figure A: The failure examples produced by our proposed method.

We thank reviewers for their comments which are very helpful for improving the paper. We address reviewers' comments
in the rebuttal and will revise the paper accordingly. All the results are evaluated by $\text{mAP}^r_{0.5}$ unless specified otherwise.
**R1:** *About the tightness assumption. How sensitive/robust is the proposed method?* As suggested, we evaluate the
proposed method by expanding/contracting the bounding boxes and show its performance under different expan-
sion/contraction ratios in following table. The tightness prior is helpful to construct the training bags, and the proposed
method is quite robust to the small ratios from $-5\%$ to $+5\%$. Moreover, by slightly contracting the bounding boxes,
e.g. $-5\%$, we can reduce noisy pixels in each positive bag, resulting in even better performance. However, excessively
expanding or contracting ratios leads to unreliable positive and negative bags, and thus produces sub-optimal results.

| Ratio | $+15\%$ | $+10\%$ | $+5\%$ | $0\%$ | $-5\%$ | $-10\%$ | $-15\%$ |
|---|---|---|---|---|---|---|---|
| Ours | 54.0 | 55.0 | 58.0 | 58.9 | 59.5 | 46.0 | 28.1 |

**R1:** *About the CVPR'18 paper "Learning ..."* We will cite it, and discuss its similarities and differences from our paper.
**R2:** *About average pooling.* Only with the unary term, the result of average pooling is 36.8, which falls behind the
result of max pooling, 43.9, because average pooling considers all pixels in the positive bags as the foreground and
overestimates the object region. Therefore, using max pooling can better diminish false alarms.
**R2:** *Result on COCO and small objects.* The result on *coco minival* is shown in following table (BoxMask is the
baseline method in the paper). Our method outperforms the baseline and reaches $78.3\%$ ($= 45.5/58.1$) and $68.3\%$
($= 11.2/16.4$) of the performance of fully-supervised Mask R-CNN in $\text{AP}^r_{0.5}$ and $\text{AP}^r_S$, respectively. The results show
that our method is effective in segmenting diverse objects with varied sizes.

| method | AP | $\text{AP}^r_{0.5}$ | $\text{AP}^r_{0.75}$ | $\text{AP}^r_S$ | $\text{AP}^r_M$ | $\text{AP}^r_L$ |
|---|---|---|---|---|---|---|
| BoxMask | 11.1 | 31.1 | 6.0 | 5.3 | 11.6 | 15.9 |
| Ours | 21.1 | 45.5 | 17.2 | 11.2 | 22.0 | 29.8 |
| Mask R-CNN [7] | 36.3 | 58.1 | 38.5 | 16.4 | 38.9 | 53.5 |

**R2:** *Failure cases.* Figure A shows failure cases of our method. In (a) and (b), segments of small objects are incomplete
due to inaccurate boundary on low-resolution regions. In (c) and (d), different instances are wrongly merged. In (e) and
(f), inaccurate object contours are segmented due to inter-instance similarity and cluttered scenes.
**R3:** *Performance versus annotation effort.* Based on [Bellver et al, Budget-aware Semi-Supervised Semantic and
Instance Segmentation, CVPR'19 workshop], the instance-level and box-level annotation costs of Pascal VOC are 239.7
and 38.1 seconds per image, respectively. We train Mask R-CNN by instance-level annotation, and limit the amount of
annotation so that the annotation budget is comparable with $1.0\times, 1.5\times, 2.0\times$ of the box-level annotation. The results
of the three different settings are 48.3, 53.5 and 59.9, respectively. The first two results fall behind our method by the
margins 10.6 and 5.4, while the last one surpasses ours by 1.0 but with $2\times$ annotation cost. Therefore, with the same
annotation cost, our method outperforms Mask R-CNN because less training data lead to overfitting for Mask R-CNN.
**R3:** *The four questions about Eq.4.* (1) Epsilon is an 8-neighbor set. (2) Because neighboring pixels are connected in
the pairwise term, we can propagate the segment scores, and thus enlarge the segment. (3) We assume "patches" in the
review refers to "bags" in our method. If all instances in positive bags are treated as positive samples, many background
pixels are mistaken as positive samples. Therefore, more false alarms could be predicted. (4) With the tightness prior,
each positive bag meets the MIL assumption, so this task could be solved with the MIL formulation.
**R3:** *Comparing efficiency with [16].* The method [16] requires pre-generated proposals from Selective Search, obtains
the detected boxes by Fast R-CNN, and finally takes the detected boxes and the RGB images as the input to generate
the instance result. In [16], the proposal generation step alone takes $\approx 10$ seconds per image. In contrast, our method
requires only one forward pass which takes only $\leq 0.1$ seconds per image or $\approx 1$ second per image if DenseCRF is
applied. Therefore, our method is more efficient than [16].
**R3:** *About L85.* We agree with this comment, and will make it more clear in the revision. However, what we want to
claim is that in general, the applicability of the fully supervised methods "may" be limited in the real world because of
the high annotation cost.
**R3:** *Comparing performance with fully supervised Mask-RCNN.* Our method adopting the MIL formulation tends to
highlight the discriminative parts of objects, while Mask R-CNN with mask-level annotation emphasizes the whole
objects. The IOUs between the ground truth and the discriminative regions are often larger than 0.25 but less than
0.5. It is why our method slightly outperforms Mask R-CNN in $\text{mAP}^r_{0.25}$, but falls behind it in $\text{mAP}^r_{0.5}$, $\text{mAP}^r_{0.7}$, and
$\text{mAP}^r_{0.75}$, as reported in Table 1 of the submitted manuscript.

[Meta-Review · NeurIPS 2019]

Although the bounding box supervision is relatively more costly than the simple image-level class labels, the proposed method exploits the benefit of the given annotation effectively and achieves good performance. In particular, the MIL formulation is interesting. I recommend accepting this paper.